# The Influence of the Atmospheric Electric Field on Soil Redox Potential

**Konstantinos Kourtidis** [1,*] and **Michel Vorenhout** [2,3]

1   Department of Environmental Engineering, Demokritus University of Thrace, 67100 Xanthi, Greece
2   Institute for Biodiversity and Ecosystem Dynamics—Freshwater and Marine Ecology (IBED-FAME),
    University of Amsterdam, P.O. Box 94240, 1090 GE Amsterdam, The Netherlands;
    m.vorenhout@mvhconsult.nl
3   MVH Consult, 2317 BD Leiden, The Netherlands
*   Correspondence: kourtidi@env.duth.gr; Tel.: +30-2541079383

**Abstract:** Atmospheric electric fields (AEFs) have recently been proposed to link to biogeochemical processes below the Earth's surface by means of a charge separation. Despite the potential importance of such a process, up to now we almost completely lack the relevant measurements. Here, we extend the database with 2 months of concurrent soil redox and atmospheric electric field measurements. It appears that the changes that occur in the order of days in soil redox are at periods anticorrelated with the logarithm of the positive values of the AEF. However, weather conditions might be driving the anticorrelation rather than a direct link, as the synoptic weather conditions appear to influence soil redox. Soil redox does not respond to changes in the AEF that are of shorter duration, either minutes or several hours, except in some cases of very negative AEFs or very high field strengths in the presence of moderate rainfall. In such a case, the variation in soil redox could be associated with a mechanism that transfers charge to the ground or brings ions towards the ground's surface. To reach firmer conclusions on the effect of the AEF on soil redox, we need to extend the range of collocated soil redox and AEF measurements so that they cover at least one year.

**Keywords:** soil redox; atmospheric electric field; potential gradient

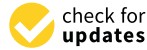



## 1. Introduction

The redox potential is a value for the ability of a medium to take up or release electrons. Soil redox potential (Eh), ranges from approximately −600 mV to +800 mV, with lower values for reducing conditions. Soil redox is influenced by the oxygen content, which, in turn, is determined by the amount of water in the soil system and the metabolic activity of microorganisms, among others.

Soil redox can be determined by a precise voltage measurement between an inert probe and a reference. While many studies have improved our understanding of the various drivers and processes governing the subsurface electrochemical environment, many of the observed variations are difficult to reconcile with known drivers of electrochemical heterogeneity, and hence, a comprehensive understanding of the variability in subsurface electrochemistry has not been reached yet. Small-scale variations in the electrochemical properties of sediments and soils are mainly controlled by biotic influences. Photosynthesis also promotes fluctuations in redox conditions by introducing oxygen into the upper layers of soils and sediment [1,2], resulting in a net diurnal increase in oxygen concentrations and a net nocturnal decrease caused by respiration. Indeed, various abiotic drivers of spatial linkages and synchronized temporal variability in subsurface chemical concentrations and microbial activity have been identified. They include solar radiation, groundwater flow, atmospheric pressure, tidal cycles, and gradients of the chemical potential of charge carriers. Despite the breadth of understanding of the processes governing the Earth's subsurface electrochemical environment and the consequences for organisms, the known drivers fail

to explain all observed electrochemical variations. This is especially true for variations in the deeper layers (up to meters) of the Earth's surface [3].

It was recently proposed that variations in the atmospheric electric field (AEF, also referred to as the potential gradient, PG) can influence the electrochemical environments of soils and water bodies and their sediments, with implications that would be relevant for a wide range of ecosystems [4,5]. A vertical AEF is present in the atmosphere even in fair weather conditions, driven by the global electric circuit (GEC) between the ground and the ionosphere, which is charged by thunderstorm regions (e.g., [6]). The AEF near the ground exhibits variations which are partly driven by the GEC and partly by local conditions such as pollution, the presence of radon, atmospheric dust, the overhead passage of charged clouds (e.g., [7]), and hydrometeors (e.g., [8]). While in fair weather conditions the AEF is usually of the order of 50–200 V/m, it can reach or exceed +/−10 kV/m during thunderstorms.

The near-ground AEF has recently been linked to biogeochemical processes below the Earth's surface by means of a charge separation between relatively negative charges in the Earth's interior and positive charges in the atmosphere [4]. Changes in atmospheric charges were observed to drive a subsurface migration of nutrients that are essential to microbial metabolism, thereby affecting the efficiency and dynamics of microbial processes [4]. If the atmospheric electric field can alter the availability of nutrients, this will affect the soil microbial activity, which, in turn, will impact soil redox.

The AEF might have an influence on soil redox conditions through a number of mechanisms. Overall, the influence of the atmospheric electric field on soil redox conditions may be complex and can depend on a variety of factors such as soil type, vegetation cover [5], ion movement, the release of respiratory electrons induced by currents caused by variations in the AEF [4], and meteorological conditions. One way that atmospheric electricity may affect soil redox potential is through the migration of ions or charged particles in the soil. The atmospheric electric field can create a potential gradient in the soil, which can cause ions to move towards or away from the soil surface. This movement of ions can affect the redox potential of the soil. For example, positively charged ions such as calcium ($Ca^{2+}$) and magnesium ($Mg^{2+}$) may be attracted towards the soil surface under the influence of the AEF, leading to increased soil alkalinity and more reducing conditions.

One of the most obvious ways that atmospheric electricity may affect soil redox potential is through lightning strikes. Lightning produces $NO_X$ in the atmosphere, which is then transferred to the soil as fixed nitrogen. It can also enhance the horizontal transfer of microbial genes [9], thus altering the microbial soil ecology. Microbes in the soil play an important role in soil redox chemistry, as they can consume or produce electron donors and acceptors. The first report to show that lightning has effects on soil properties, organic matter availability and microecology, and plant growth has been published very recently [10]. Also, very recent results show that lightning can cause phosphorous reduction [11] and other redox changes [12,13] in soils.

Overall, while the atmospheric electric field may not be the primary factor affecting soil redox potential, it can play a role in shaping the redox chemistry of soil through several different mechanisms.

More research is needed to fully understand the mechanisms by which the atmospheric electric field affects soil redox conditions. However, the available data from which the above inferences are deduced remain very sparse. Hence, here we present concurrent AEF, soil redox, and meteorological parameter measurements which might add new insights. The presented two-month dataset of co-located concurrent measurements of the AEF and soil redox is the longest reported up to now. The concurrent meteorological parameter measurements are also new, as well as the fact that soil redox has been measured concurrently in different depths in the soil.

## 2. Materials and Methods

Measurements were performed at the Xanthi site from 27 October 2017 until 3 January 2018. Hence, they consist of the largest dataset of concurrent PG and soil Eh measurements presented to date. Eh was measured at depths of 1 cm and 20–21 cm from the surface. The site (41.15° N, 24.92° E, 75 m above sea level) is a rural site on the Campus of Democritus University of Thrace near the town of Xanthi (population around 65,000). The measurement site is flat and is seated at the edge of a smooth, south-facing slope. The ground surface in the immediate vicinity of the station is covered with soil and grass. More details can be found in [14].

PG measurements have been performed since 2011 at the site [14,15]. PG was measured at 1 s intervals with a Campbell CS110 electric field meter. This device has an internal CR1000M datalogger that is linked via an RS232-Ethernet convertor to a locally running Loggernet setup.

A multichannel datalogger (HYPNOS IV, MVH Consult, Leiden, The Netherlands) was connected to probes for the continuous monitoring of Eh in the soil. Four sturdy probes (Paleo Terra, Amsterdam, The Netherlands) with redox measurement tips at −1 and −21 cm below the surface were placed in a created mud pit. Installation directly into the soil proved impossible due to the very low permeability of the soil. The mud pit was dug 1.5 m from the fence around the local weather station/PG meter.

The Hypnos datalogger is field-deployable, relatively cheap, and runs autonomously on batteries. The logger has an impedance for each separated redox channel of over 10 TOhm, providing extreme stable measurement conditions and removing any drift from the measurements. More details on the datalogger and the probes can be found in [3].

## 3. Results and Discussion

There is emerging evidence of some form of coupling between atmospheric electricity at various scales and various biological processes in land environments [16], and references therein. It has been proposed that variations in atmospheric electricity can influence the electrochemical environments of soils [4]. Hunting et al. [5] reported 5 days of soil redox and positive atmospheric ion measurements, and noted that both have distinct diel cycles that appear to co-vary. Hunting et al. [5] presented a figure depicting eight days of simultaneous measurements of soil redox, PG, and meteorological variables, and noted that diurnal cycles in soil redox potential did "not seem to be directly governed by changes in the atmospheric potential gradient".

Here, a much longer dataset is presented. The dataset of collocated potential gradient and soil redox measurements at 1 cm and 20 cm depth is around 70 days (Figure 1).

One interesting feature that becomes apparent through the use of the logarithmic scale is that the long-term (in the order of days) changes in soil redox appear at periods anticorrelated with the logarithm (base 10) of the atmospheric potential gradient (Figure 1). Such periods are at hours 240 to 400, 600 to 840, and 132 to 1560 (Figure 1). In Figure 1, only positive PG values are displayed. Negative value excursions appear only in disturbed weather conditions, for example, around rain events, the passage of very electrified low stratiform clouds, or lightning [17]. The reason for the exclusion of negative PG values from Figure 1 is the need to use the logarithmic scale for PG. This is also justified by the very short duration of the negative excursions.

Below, whether PG and soil redox appear to co-vary in shorter time scales, of the order of minutes, is examined.

On 6 November (Figure 2a), there was no rain and additionally PG exhibited its typical fair weather diurnal variation for the site [14,15]. The diurnal pattern of soil redox potential was such that it exhibited lower values during local noon (10:00 UTC, in the figure) when PG was at its diurnal maximum, at around 200 V/m. The observed variations in the PG, of the order of +100 V/m increases or −100 V/m decreases occurring over a period of a few minutes, did not seem to influence soil redox.

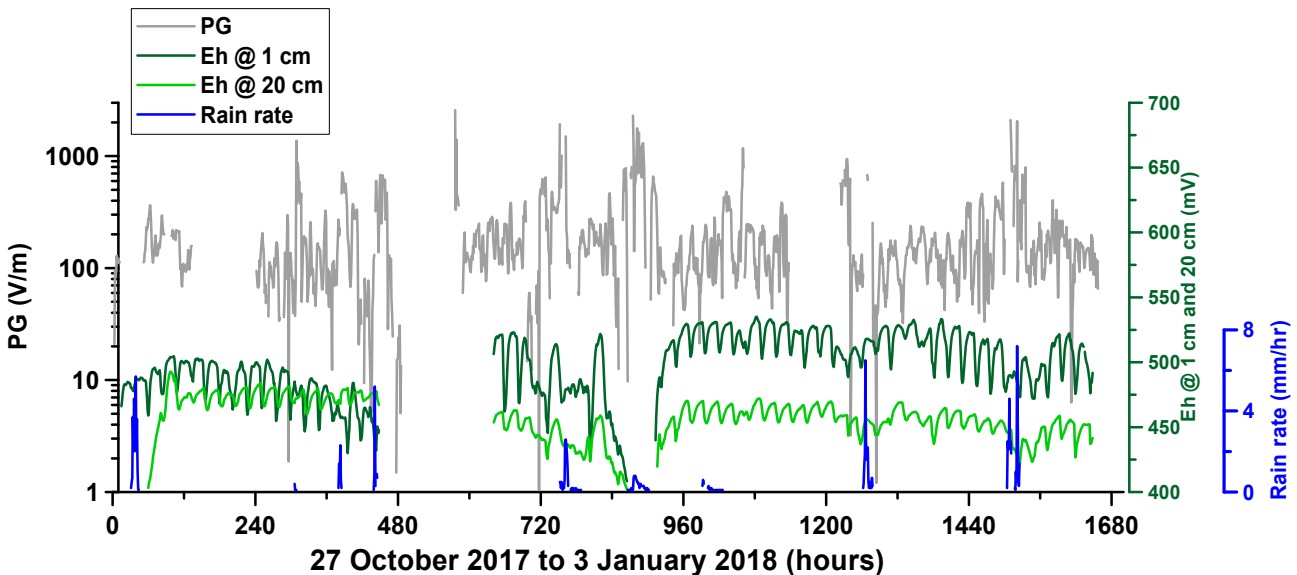

**Figure 1.** The timeseries of collocated potential gradient and soil redox measurements. Soil redox measurements at 1 cm and 20 cm depth are shown. Hourly mean values are shown. Only positive PG values are shown.

On 25 December (Figure 2b), there was no rain, but the afternoon values of PG were disturbed and reached 400 V/m, even exceeding the midday maximum of 200 V/m. This could be due to the passage of charged clouds overhead. The observed short term variations in PG, of the order of +/−100 V/m, again did not seem to influence soil redox.

On 8 December (Figure 2c) there was very light rain throughout the day. The observed short-term variations in PG, which on this day were of the order of +/−300 V/m, again did not seem to influence soil redox.

Hence, it appears that, on days with no rain or light rain, the 1 min variations in PG did not have any effect on soil redox (Figure 2a–c). Even when the PG changed by 300 V/m within 1 min (Figure 2c), soil redox values stayed unaffected.

On 28 December 2017 (Figure 2d), there was moderate rain in the afternoon after 14:00 UTC (the local time is UTC+2). On this day, there were some variations in soil redox that were concurrent and in the same direction as those of the PG; for example, at around 15:00, in 10 min the PG dropped by 2200 V/m and soil redox at 1 cm depth dropped by 8 mV, while soil redox at 21 cm depth remained unaffected (Figure 2d). It does not seem very probable that, at that moment, lightning struck near the site and caused the disturbance of soil redox at 1 cm depth. The community-based lightning detection network blitzortung.org shows no lightning activity in the area for this day. One should keep in mind though that the detection efficiency of such networks is not 100% and could at times be below 50%. Apart from lightning, any other mechanism that would transfer charge to the ground or increase the atmospheric negative ion concentration near the ground could have been responsible, as [4] observed direct responses of Eh to changes in atmospheric ions in laboratory experiments. Namely, [4] observed increases in Eh concurrent with increases in atmospheric positive ion concentrations near the ground. As negative PG values are associated with downward fluxes of negative ions, and hence increases in negative ions near the ground, this mechanism could be responsible for the drop in soil redox at around 15:00, as at that time the PG dropped to −2570 V/m (Figure 2d). On 29 December 2017 (Figure 2e), there was moderate rain in the early morning hours and until around 12:00 UTC. As with 28 December, there were some variations in soil redox during strong PG disturbances at around 11:00, although at around 07:00, when PG was also strongly disturbed, there was no discernible effect on the soil redox values. However, on 29 December, unlike the 28th, soil redox was disturbed at both 1 cm and 21 cm depth. Also, the observed disturbance on the 29th occurred more slowly than the one on the 28th. Any mechanism transferring charge to

the ground, such as a partial discharge, could have resulted in the observed disturbance of soil redox at both depths. Partial discharges (also known as corona discharges) are localized air breakdowns that can occur in the presence of moisture when the voltage gradient is high enough and allows charge to leak through, especially around non-insulating pointed materials, such as metal rods. This could be supported by the high PG values at the time. With 1 min PG values being around $+/-10$ kV/m, short-time PG values could easily exceed that by more than an order of magnitude. As the relative humidity was 90–100% on that day, partial discharges could occur more easily. Metal objects in the vicinity, such as the meteorological mast, or the fence surrounding it, near which the soil redox measurements were made, could also facilitate this. The observed effects on soil redox during both days did not last very long; it appears that they lasted for around one hour.

In Figure 2f, the mean PG and mean Eh at 1 cm and 21 cm depth are displayed per circulation weather type (CT). CTs were computed for each day. Here, we used a PXE (Pca-eXtreme scores reassigned using Euclidean distance) principal component analysis (PCA) categorization of 10 CT classes. The PXE was created by using the cost733class software [18] and reanalysis data from ERA-Interim [19]. The daily mean sea-level pressure, the 500 hPa geopotential height, the thickness of the layer between the 850 hPa and 500 hPa isobaric surfaces, and the relative vorticity at 500 hPa were used in the input and classified. Briefly, CT1and CT4 are characterized by barometric lows over Europe which result in W-SW surface flow over the site. CT2 is associated with a high over the Balkans and results in NW flow over the site. CT3 is characterized by S surface flow over the site which can result in Saharan dust transported to the site. Although this is a CT with very low lightning frequency, it is the CT with the highest occurrence of large positive PG values. In CT5, lows exist over the Middle East and N Atlantic and a high over S Russia. The height of the 500 hPa isobaric and the thickness of the layer between the 850 and 500 hPa isobaric surfaces decrease gradually from N Africa to N Europe, while vorticity has negative values over most of Europe. In CT8, lightning activity over continental Europe, the Balkans, and the Eastern Mediterranean is high. CT10 is characterized by E-NE surface flow over the site, high positive vorticity at 500 hPa, a low 500 hPa isobaric surface, and a thin 850–500 hPa thickness, as well as steep gradients of the latter two. The high positive vorticity over the Balkans and Eastern Mediterranean means that there is considerable updraft during CT10. Temperature and humidity are low in Xanthi while rain and wind are high during CT10. More information on the computation method and more information on the type of weather conditions each CT is associated with is provided in [20]. Circulation weather types 6, 7, and 9 were not encountered during the measurement period, as they are very uncommon in winter [20]. The daily means for PG and Eh were computed from the hourly data, and the mean of the daily means per CT was computed. It appears that the weather types leading to higher PG also lead to lower Eh, especially at 1 cm depth (Figure 2f). Hence, the apparent anticorrelation of PG and soil redox in Figure 1 might be due to meteorological reasons, i.e., the prevailing synoptic weather conditions might lead to anticorrelation rather than a direct link between PG and Eh.

These results cannot be directly compared with Hunting et al.'s [5] measurements of eight days of simultaneous measurements of soil redox and PG, because in the latter most days, soil redox did not have a distinct diel cycle, and also, the PG values were much higher than the ones in the present study. In [4], it was reported that Eh, in the shallow littoral zone of a lake in the Swiss Alps, had a diel cycle similar to the one of the GEC. In [4], for the site mentioned above, the minimum in Eh was observed at around 8:00 UTC (10:00 local time) and the maximum at around 21:00 UTC (23:00 local time); in a ditch in the Netherlands, the maximum was observed at around 13:00 UTC (15:00 local time) [4], while in Xanthi, a distinct minimum was observed at around 10:00–12:00 UTC (13:00–15:00 local time). It is difficult to directly compare Eh results between the Swiss lake sediment and the Xanthi soil.

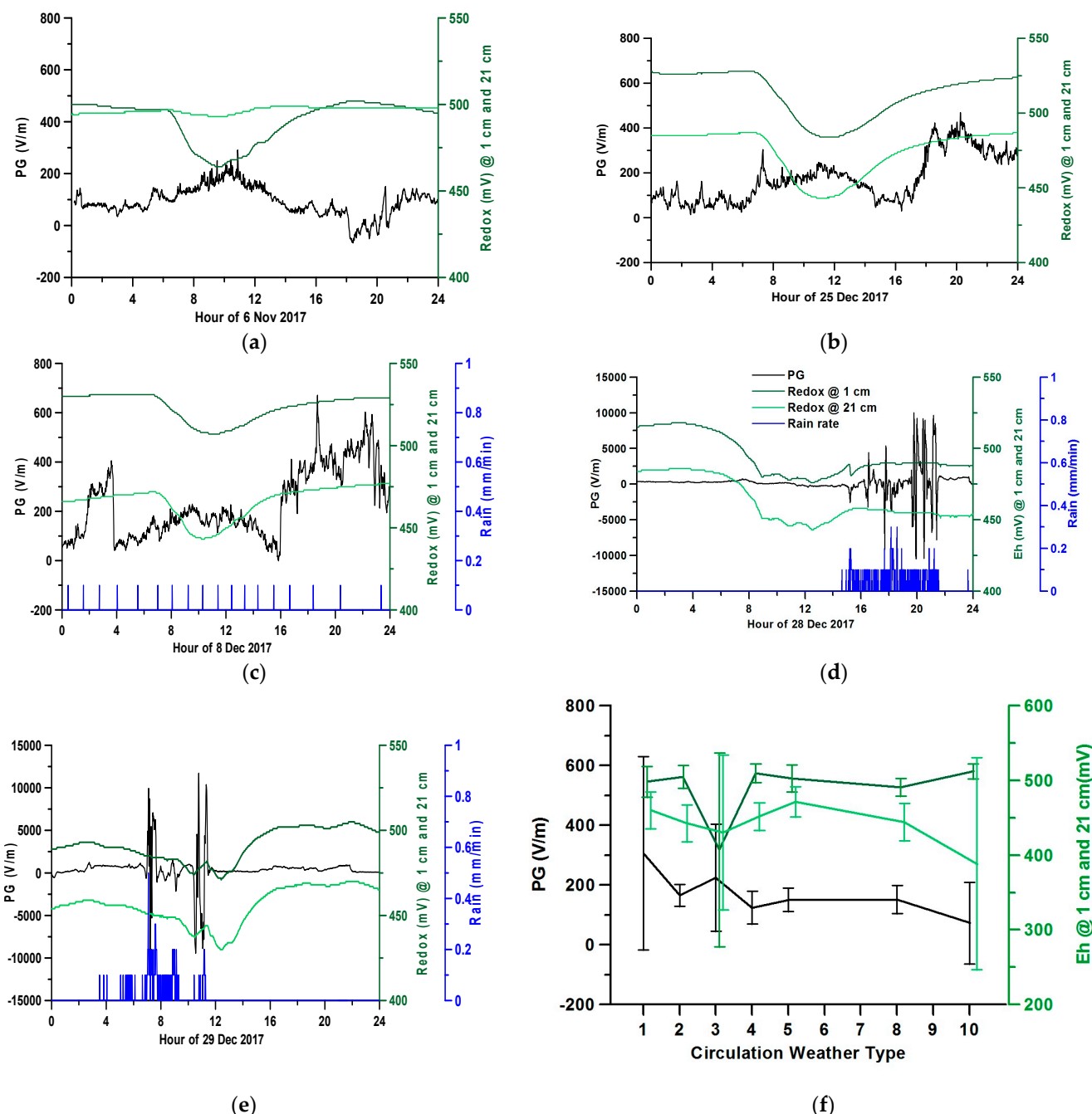

**Figure 2.** Atmospheric potential gradient (black), soil redox at 1 cm (dark green) depth, 21 cm (light green) depth, and rain rate (blue) (**a**) on 6 November 2017 and (**b**) on 25 December 2017, two days with no rain, (**c**) on 8 December 2017, a day with light rain, (**d**) on 28 December 2017, and (**e**) on 28 December 2017, two days with moderate rain. The time resolution of the measurements is 1 min. Time is in UTC. Local time is UTC+2. (**f**) Mean PG and soil redox during different circulation weather types. The vertical bars are +/−1 σ (standard deviation) of the daily means. The soil redox values are shifted slightly to the right so that the standard deviation bars do not overlap. Note the different potential gradient scales of figures (**d**,**e**).

## 4. Conclusions

It appears that the changes that occur in the order of days in soil redox are at periods anticorrelated with the logarithm of the positive values of the atmospheric electric field; however, meteorological conditions might also be driving the anticorrelation instead of a direct link between the AEF and Eh, as it appears that the synoptic weather conditions

influence Eh. Soil redox does not respond to changes in the AEF that are of shorter duration, either minutes or several hours, except in some cases of either large drops in the AEF to very negative values or very high absolute AEF values in the presence of moderate rainfall. In such a case, the variation in soil redox could be associated with a mechanism that transfers charges to the ground, such as a partial discharge, or transports ions near the ground, although the available data do not allow for firmer conclusions at the moment.

To reach firmer conclusions on the effect of the AEF on soil redox, the duration of collocated soil redox and AEF measurements need to be extended so that they cover at least one year. It would be also worthwhile to have a lightning detector as well as the measurements of atmospheric ions near the ground at the site so that one can reach definite conclusions about the effects of discharges and atmospheric ions on soil redox.

**Author Contributions:** Conceptualization, K.K. and M.V.; methodology, K.K. and M.V.; formal analysis, K.K. and M.V.; investigation, K.K. and M.V.; resources, K.K. and M.V.; data curation, K.K. and M.V.; writing—original draft preparation, K.K.; writing—review and editing, K.K.; visualization, K.K.; funding acquisition, K.K. All authors have read and agreed to the published version of the manuscript.

**Funding:** One of us (M.V.) received financial support for a short-term scientific mission to the Xanthi site from COST Action CA15211 "Atmospheric Electricity Network: coupling with the Earth System, climate and biological systems", supported by the COST (European Cooperation in Science and Technology) Program, EU Framework Programme Horizon 2020.

**Institutional Review Board Statement:** Not applicable.

**Informed Consent Statement:** Not applicable.

**Data Availability Statement:** The data presented in this study are available on request from the corresponding author.

**Acknowledgments:** We acknowledge the help of A. Karagioras with the PG and meteorological data. We also acknowledge A. Nita for the provision of the circulation weather type data. This work was initiated based upon discussions and interactions with COST Action CA15211 "Atmospheric Electricity Network: coupling with the Earth System, climate and biological systems", supported by the COST (European Cooperation in Science and Technology) Program.

**Conflicts of Interest:** The authors declare no conflict of interest. The funders had no role in the design of the study; in the collection, analyses, or interpretation of data; in the writing of the manuscript; or in the decision to publish the results.

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
