# Peer review of "The Influence of the Atmospheric Electric Field on Soil Redox Potential"

_oxygen, doi:10.3390/oxygen3040025_

Round 1

Reviewer 1 Report (Previous Reviewer 1)

The authors had revised the manuscript according to the review's suggestions. I think the manuscript is acceptable within the scope of Oxygen.

 Minor editing of English language required

Author Response

We made some minor editing of the English language.

Reviewer 2 Report (Previous Reviewer 2)

Using observed data, the authors described the relations between atmospheric electric field and soil redox. And they tried to explain the relations in the manuscript. I think the lightning data would be crucial to support the manuscript's results. However, the lightning data was not available in the study. I think that the authors had better analyze the more detail about the relations between rainfall and electric field and soil redox. Therefore, I would like to recommend this manuscript to be accepted after major revision.

Specific comments:

1. Lines 129-135: The authors compared their values before/after and during the rainfall period.

2. Lines 166-167: The authors need to change the paragraph because the lightning data was not available.

3. Lines 188-202: The authors classified circulation weather types in this paragraph. I think that the authors need to explain the circulation weather types in more detail like what the number does mean. 

4. Lines 217-224: The authors need to change the word "meteorology" to "rainfall" in order to specify the results. Because the authors used only rainfall data among meteorological elements.

Author Response

We thank the reviewer for the comments, to which we respond below.

"Using observed data, the authors described the relations between atmospheric electric field and soil redox. [...] I think the lightning data would be crucial to support the manuscript's results. However, the lightning data was not available in the study. I think that the authors had better analyze the more detail about the relations between rainfall and electric field and soil redox": We reevaluated thoroughly our hypothesis about lightning, which we now tend to dismiss. We offer now alternative explanations for the observed relationships. Please see the revised manuscript.

Response to specific comments below.

"Lines 129-135: The authors compared their values before/after and during the rainfall period": We are not sure we understand this comment correctly. For more details on the complex relationship of rain and AEF, please see refs [8], [20] of the manuscript.

"Lines 166-167: The authors need to change the paragraph because the lightning data was not available": In the revised version we address this and drop the lightning hypothesis. Please see the revised manuscript.

"Lines 188-202: The authors classified circulation weather types in this paragraph. I think that the authors need to explain the circulation weather types in more detail like what the number does mean": The number refers to the numbering of CT classification. In the classification we used there are 10 different CTs, numbered from 1 to 10. The reader is referred to [18], [19] and [20] for more details on CTs, as already mentioned in the manuscript.  

"Lines 217-224: The authors need to change the word "meteorology" to "rainfall" in order to specify the results. Because the authors used only rainfall data among meteorological elements": Meteorology is correct here. We refer to the circulation weather types, which is not rainfall. CTs are a way of categorizing synoptic weather types. A CT encompasses in its calculation many different aspects of weather and different meteorological variables.

Reviewer 3 Report (New Reviewer)

The manuscript „On the influence of the atmospheric electric field on soil redox Potential“ describes and analyse 2 months of measurements of concurrent soil redox and atmospheric electric field measurements. Based on the measurements findings are made mainly in the form of hypotheses.  

My major point is that the authors point out that they did not have lightning measurements and yet the potential influence of lightning appears in their conclusions/hypotheses. Lightning activity information can be obtained from various networks. Why was this information not used?

Word “anticorrelation” is used several times. What do you mean by that? Could you calculate some type of correlation coefficient and discuss their values?

The conclusions derived from the measurements are too hypothetical in my opinion, and at least some of them should have a more solid basis (e.g. the effect of lightning). From this point of view, the article is rather technical and not a research article.

Specific comments:

What is meant by meteorology or meteorological reasons? Please say it more specifically.

Could you describe Xanthi site in more details, e.g. coordinates, type of land?

L221 – “heavy moderate rainfall” is not meaningful expression

Author Response

Below our response to reviewer number 3, which we thank for the comments.

"Based on the measurements findings are made mainly in the form of hypotheses": That the findings are mainly hypothesis is not true. The only hypothesis was about a possibility of lightning influencing soil redox, which we now revised (please see also below). 

"My major point is that the authors point out that they did not have lightning measurements and yet the potential influence of lightning appears in their conclusions/hypotheses. Lightning activity information can be obtained from various networks. Why was this information not used?": It is true that there is lightning activity from a couple of networks, although their detection efficiency, especially for low peak currents can be at times quite low (a lot gets undetected). We have now checked with one such network, and as we also now tend to dismiss the lightning hypothesis, text has been changed to: “On the 28th of December 2017 (Figure 2d) ……. On this day, there were some variations in soil redox that were concurrent and in the same direction as those of PG, as for example at around 15:00, where in 10 minutes, PG dropped by 2200 V/m and soil redox at 1 cm depth dropped by 8 mV, while soil redox at 21 cm depth remained unaffected (Figure 2d). It seems not very probable that at that moment, lightning has struck near the site and caused the disturbance of soil redox at 1 cm depth. The community-based lightning detection network blitzortung.org shows no lightning activity in the area for this day. One should keep in mind though that as the detection efficiency of such networks is not 100% and could at times be below 50%. Apart from lightning, any other mechanism that would transfer charge to the ground or increase the atmospheric negative ions concentrations near ground could have been responsible, as [4] observed in laboratory experiments direct responses of Eh to changes in atmospheric ions. Namely, [4] observed increases in Eh concurrent with increases in atmospheric positive ion concentrations, and as negative PG values are associated with downward fluxes of negative charge carriers, this mechanism could be responsible for the drop in soil redox at around 15:00, as at that time the PG dropped to -2570 V/m. On the 29th of December 2017 (Figure 2e) …. However, on the 29th of December, unlike the 28th, soil redox was disturbed at both 1 cm and 21 cm depth. Also, the observed disturbance on the 29th occured more slowly than the one on the 28th. Any mechanism transferring charge to the ground, such as a partial discharge, could have resulted in the observed disturbance of soil redox at both depths. Partial, or corona, discharges are localized air breakdowns that can occur at the presence of moisture even on insulating surfaces. This could be supported from the high PG values at the time. With 1-min PG values being around +/-10 kV/m, short-time PG values could easily exceed that by more than an order of magnitude. As relative humidity was 90-100% at that day, partial discharges could occur more easily. Metal objects in the vicinity, such as the meteorological mast, or the fence surrounding it, near which the soil redox measurements were made, could also facilitate this.

"Word anticorrelation is used several times. What do you mean by that? Could you calculate some type of correlation coefficient and discuss their values?": Anticorrelation is used 3-4 times, all times with regard to the apparent anticorrelation of soil redox with the log of AEF in Fig. 1. Yes, we can calculate e.g. linear regression, but no meaningful interpretation can be made of the low r2, since (as the reviewer can see in Fig. 1) the daily variation of AEF is very large (factor of 2-4 changes from daily min to daily max) and causes the data points to spread.  

"The conclusions derived from the measurements are too hypothetical in my opinion, and at least some of them should have a more solid basis (e.g. the effect of lightning)": Manuscript revised accordingly. See also above, as well as the revised manuscript.

"What is meant by meteorology or meteorological reasons? Please say it more specifically": We mean the prevailing synoptic meteorological conditions, i.e. the weather. We rephrased certain parts of the manuscript accordingly.

"Could you describe Xanthi site in more details, e.g. coordinates, type of land?": Description added as follows: “The site (41.15° N, 24.92° E, 75 m above sea level)is a rural site on the Campus of Democritus University of Thrace  near the town of Xanthi (population around 65,000). The measurement site is flat and is seated at the edge of a smooth, south-facing slope. The ground surface in the immediate vicinity of the station is covered with soil and grass. More details can be found in [14].”  

"L221 – “heavy moderate rainfall” is not meaningful expression": Indeed it is not meaningful. This was a typos, we meant heavy to moderate. We now corrected to simply “moderate rainfall”. The maximum rainfall during the 28th was 4.6 mm/hr and the 29th of December was 7.2 mm/hr. Strictly speaking, the USGS considers heavy rain as rain between 4 and 8 mm/hr, the World Meteorological Organistation as rainfall >= 50 mm/day, other agencies rainfall > 7.6 mm/hr, so there is not a very strict definition. Anyway, we adopted “moderate rainfall” in the text since the daily rain amount did not exceed the 50 mm/day of the WMO definition.

Round 2

Reviewer 2 Report (Previous Reviewer 2)

I think the authors tried modifying the manuscript according to the comments. I would like to recommend the manuscript would be accepted after minor revision. However, there is some misunderstanding of my previous comments. 

1. About Figure 1: The authors need to zoom out the period of rainfall so that the readers can understand the anticorrelation. And authors had better describe the values of PG before, after, and during the rainfall in the manuscript.

2. About CT: I think that the readers can refer to the previous manuscript. However, I think that the variarion of PG with CT is one important result of the manuscript. Therefore, I would like to recommend that the authors need to describe the meaning of the number in the manuscript. For example, 1 is low pressure type, 2 is frontal type, etc..

Author Response

"About Figure 1: The authors need to zoom out the period of rainfall so that the readers can understand the anticorrelation. And authors had better describe the values of PG before, after, and during the rainfall in the manuscript":

We are not completely sure that we understand correctly what the reviewer means by zooming out the period of rainfall. Figure 1 modified the way we understand the comment. We lowered the scale of rainfall so that the rainfall curve does not intervene much with the PG and redox curves. Regarding the description of values of PG before, after and during the rainfall, we can say that the values of PG before, after and during the rainfall can be seen very clearly at figures 2(c), (d) and (e). It is evident from these figures that the behavior of PG is complicated and not uniform during all the events. It is also clear from these figures that even the meaning of “during the rainfall” is not so clear. In figure 2(c) for example, it can be seen that it rained for 1-2 min per hour, so it rained sporadically for the whole day. The relationship of rain and PG is quite complex and out of the scope of the present manuscript. See also refs [8], [20].

“About CT: I think that the readers can refer to the previous manuscript. However, I think that the variation of PG with CT is one important result of the manuscript. Therefore, I would like to recommend that the authors need to describe the meaning of the number in the manuscript. For example, 1 is low pressure type, 2 is frontal type, etc..”:

We added a short description as follows: “CT1and CT4 are characterized by barometric lows over Europe which result in W-SW surface flow over the site. CT2 is associated with a high over the Balkans and results in NW flow over the site. CT3 is characterized by S surface flow over the site which can result in Saharan dust transported to the site. Although this is a CT with very low lightning frequency, it is the CT with the highest occurrence of large positive PG values.  In CT5, lows exist over the Middle East and N Atlantic and a high over S Russia. The height of the 500 hPa isobaric and the thickness of the layer between the 850 and 500 hPa isobaric surfaces decrease gradually from N Africa to N Europe, while vorticity has negative values over most of Europe. In CT8, lightning activity over continental Europe, the Balkans, and Eastern Mediterranean is high. CT10 is characterized by E-NE surface flow over the site, high positive vorticity at 500 hPa, a low 500 hPa isobaric surface, and a thin 850–500 hPa thickness as well as steep gradients of the latter two. The high positive vorticity over the Balkans, and Eastern Mediterranean means that there is considerable updraft during CT10. Temperature and humidity are low at Xanthi while rain and wind are high during CT10”. For more information the reader is referred to reference [20]. We also added a sentence “the synoptic weather conditions appear to influence soil redox” in the abstract, as the reviewer pointed out the importance of this result.

Reviewer 3 Report (New Reviewer)

The manuscript was improved. I have only some additional comments.

Specific comments:

L15 – I would recommend using weather condition instead of weather only.

Line 181 – Please reformulate the sentence beginning “Namely”. It is very long and difficult to follow.

Line 206 – I do not understand the beginning of the sentence “Partial, or corona, discharges are localized air breakdowns”

L246 – I would recommend “meteorological conditions” instead of meteorology.

L246 – I am not sure if I understand what you mean by electric field. Electric field is a vector with positive or negative values. How do you apply logarithm?

Author Response

“L15 – I would recommend using weather condition instead of weather only”: Done.

“Line 181 – Please reformulate the sentence beginning “Namely”. It is very long and difficult to follow”:

Sentence rephrased and split in two sentences.

"Line 206 – I do not understand the beginning of the sentence “Partial, or corona, discharges are localized air breakdowns”:

Sentence rephrased to “Partial discharges (known also as corona discharges) are localized air breakdowns that can occur at the presence of moisture when the voltage gradient is high enough, and allow charge to leak through, especially around non-insulating pointed materials, such as metal rods”.

“L246 – I would recommend “meteorological conditions” instead of meteorology”:

Done.

"L246 – I am not sure if I understand what you mean by electric field. Electric field is a vector with positive or negative values. How do you apply logarithm?":

We use only the positive values of the field. Most of the time, the AEF is positive and positive is also the mean field, at around 100 V/m. In fair weather conditions the field is also positive. Only in disturbed weather conditions (i.e. during rain, hailstorms, passage of low clouds which are very electrified or lightning) it exhibits negative excursions. We revised the text of the manuscript for clarity on this point.

This manuscript is a resubmission of an earlier submission. The following is a list of the peer review reports and author responses from that submission.

Round 1

Reviewer 1 Report

The manuscript "On the influence of the atmospheric electric field on soil redox potential" is mainly a descriptive study, although it has provided comprehensive information on atmospheric electric fields and soil redox, the authors described the phenomenon more than the in-depth discussion.

General Comments:

Line 24: Minor values should be preceded, changed to “-600 mV to +800 mV”.

Line 31: Observed?

Line 64: You use AEF to represent the atmospheric electric field at the beginning, try to use a scientific style throughout the manuscript.

Line 118: What’s the meaning of “references therein”?

Line 121-124: Just discuss the difference between Hunting et al. And yours, not include so many details.

Line 125: Deleted "We will present here a much longer dataset. ".

Line 133-134: Rephrase the text (We will now … ) according to a scientific writing style. Try to use a scientific style throughout the manuscript. 

Line 150: You used two different colors to represent the soil redox at 1 and 20 cm depth, there is no need to use two Y-axes, just add a figure legend.

Line 184: It’s more clear to add a figure legend for all.

Line 151-183: The results should be written in the past tense.

Extensive editing of English language required

Reviewer 2 Report

The manuscript deals with soil redox potential considering atmospheric electric field during rainfall and lightning. The authors used the experimental data obtained from observation and analyzed them. However, I think the authors need to add more detailed descriptions to support their results. And I could not find out the conclusions in the current manuscript. Therefore, I would like to recommend that it would be accepted after major revision.

Comments

1.  Line 15: PG would be full spelling.

2. Lines 130-132: Authors described "Nagative values appeared only around heavy rain" in the manuscript. I found that the maximum rainfall rate would be 8 mm/hr. Is this rainrate defined heavy rainfall? Authors need to describe the time when data is not available.

3.  Figure 2 : How the authors could explain the reduction of soil redox at the time 08 to 12 LT. And the authors need to show the meaning of each line.

5. Line 158: How do authors define the heavy rainfall.

6. Line 163: How do we know the lightning was occurred at that time?

7. Authors need to add Conclusions section in the manuscript.

Reviewer 3 Report

The article carried out a preliminary study on the influence of the atmospheric electric field on soil redox potential.

The question is quite interesting and relevant. Unfortunately, this article has not been studied enough. The data series are too short for any valid conclusions. Figure 1 in any case requires significant revision. The authors are encouraged to conduct additional research.